# Macroporous Oil-Sorbents with a High Absorption Capacity and High-Temperature Tolerance Prepared Through Cryo-Polymerization

**DOI:** 10.3390/polym11101620

**Published:** 2019-10-07

**Authors:** Abdul Haleem, Jia-Yun Wang, Hui-Juan Li, Chuan-Shan Hu, Xi-Chuan Li, Wei-Dong He

**Affiliations:** CAS Key Laboratory of Soft Matter Chemistry, Department of Polymer Science and Engineering, University of Science and Technology of China, Hefei, Anhui 230026, China; haleem@mail.ustc.edu.cn (A.H.); wjy0526@mail.ustc.edu.cn (J.-Y.W.); lhj2015@mail.ustc.edu.cn (H.-J.L.); hucs@mail.ustc.edu.cn (C.-S.H.); xcli@mail.ustc.edu.cn (X.-C.L.)

**Keywords:** oil clean-up, macro-porous material, hydrophobic gel, oil-sorbent, cryo-polymerization

## Abstract

The facile preparation and admirable performance of macro-porous poly(lauryl acrylate)-based oil-sorbents for organic solvents and oils are reported in this manuscript. Cryo-polymerizations of lauryl acrylate (LA) with ethylene glycol dimethacrylate (EGDMA) as the cross-linker were carried out at temperatures below the freezing point of the polymerization mixture. The polymerization medium and pore-forming agent was 1,4-dioxane. The influences of the total monomer concentration, EGDMA content and cryo-polymerization temperature on the structure of the obtained P(LA-*co*-EGDMA) cryogels were investigated with the techniques of Fourier transform infrared spectroscopy, scanning electron microscopy, contact angle measurement and thermo-gravimetric analysis. Through the modulation of the crosslinking density and porosity of these cryogels, the P(LA-*co*-EGDMA) oil-sorbents demonstrated a high absorption capacity for organic solvents and oils, recyclability and high-temperature tolerance. The absorption capacity reached 20–21 and 16–17 g/g for toluene and gasoline oil, respectively. Those fabricated sorbents survived high temperatures up to 150 °C without any change in absorption capacity as well as porosity. Considering the convenient synthesis process and absorption performance, the present work offers a remarkable opportunity to bring polymer cryogels to practical application in waste oil clean-up.

## 1. Introduction

In the modern world, there are numerous organic solvents and oils which are used across various human activities. Now and then, the discharge and leakage of these hydrophobic hydrocarbons, even including oil spilling accidents, occurs, causing great environmental problems for human beings and significant damage to wildlife [1,2,3,4,5]. In this context, oil spilling is a serious environmental disaster, causing harmful effects to terrestrial and aquatic ecosystems and yielding significant economic losses. The pollution caused by waste organic compounds, the accidental seepage of hydrocarbons and oil spilling accidents has attracted worldwide attention. Therefore, different technologies and materials have been developed and are being developed to clean up undesirable hydrocarbons [3,6,7,8,9].

The routes to cleaning waste hydrocarbons and oil spills can be catalogued into primary, secondary and tertiary treatments based on the process stage [6]. The related technologies can be divided into four categories: (1) the mechanical methods physically remove the waste hydrocarbons and oil spills with the aid of equipment such as booms, skimmers and vacuum units. They are appropriate for large-scale cleanup, but residues are usually left. (2) Chemical methods, including in-situ burning and the usage of chemical dispersants and solidifiers, are frequently utilized together with other methods because of their limitations [3,10,11]. (3) Sorbents are more widely available to clean up small areas and/or the final traces of waste hydrocarbons in water and on land. They are hydrophobic and oleophilic materials with a porous interior. The soaked sorbents can be re-used after squeezing out the absorbate. (4) Biological technologies, taking advantage of natural microbial biodegradation to detoxify or remove hydrocarbon pollutants, are considered environmentally friendly and economical [12,13]. Phytoremediation, as one kind of bioremediation, adopts the ability of plants to remove and degrade a large array of contaminants [14]. The long duration and necessary nutrients are the key problems for bioremediation methods.

Currently, there are a wide variety of sorbents used to clean up waste hydrocarbons and oil spills. Inorganic mineral sorbents such as zeolites, clays, chalk and silica gels are always hydrophobically modified with stearate, surfactant, silicone and alkylchlorosilane to attain a high absorption capacity. However, those inorganic sorbents are highly dense and expensive due to their modification cost [10]. Natural organic sorbents have the features of resource abundance, biodegradability and further enhancement of hydrophobicity [8,15,16]. In converting various natural resources into practical sorbents, a manufacture and forming process is necessary. Synthetic polymeric sorbents such as polyurethane foams and polypropylene skeins are also widely investigated and applied due to their simple preparation, the convenient adjustment of their performance and their high absorption capacity [6,8,17,18]. Carbon-based sorbents could be considered as the fourth family since they are not only made of traditional carbon materials (charcoal, activated carbon and exfoliated graphite) and carbon nano-materials (carbon nanotube and graphene) but also made from the pyrolysis of biomasses and polymers to form carbon aerogels [19,20,21]. According to the previous literature, excellent sorbents for oil cleanup should have the structural characteristics of hydrophobicity, oleophilicity, macro-porosity and reliability (low-cost, easy to fabricate and reusable).

Synthetic polymeric candidates are promising sorbents in the clean-up of waste hydrocarbons and oil spills due to the following aspects. There are many commercial hydrophobic monomers, such as lauryl acrylate (LA), and polymerization techniques to modulate the chemical structure and porous morphology of sorbents [22,23,24,25,26]. Polymerization techniques including polymerization-induced phase inversion with/without porogen [27], high internal phase emulsion polymerization [28], and cryo-polymerization [29] are frequently and effectively used to modulate the porous morphology. Physical techniques such as electron spinning [30], freezing dryness [31], and the supercritical CO_2_ foaming of polymer [32] also enjoy application in oil-sorbents. The raw materials are low-cost, the fabrication procedures are easy, and the products are available for structural in-situ modulation and post-modification. These advantages have attracted increasing attention to the development of synthetic oil-sorbents from various monomers by alternating polymerization techniques.

Cryo-polymerization has attracted much attention very recently as it is a powerful tool to produce inter-connected macro-porous materials as valuable absorbents with a rapid rate and high capacity. Principally, cryo-polymerization is carried out at temperatures below that of the polymerization mixture. During the cryo-polymerization, the frozen polymerization medium acts as the porogen, while unfrozen liquid microphase (UFLMP) is the polymerization locus to construct the framework of the obtained cryogel [33,34]. The volume fraction of UFLMP is about 0.1, causing the so-called cryo-concentration effect [35]. Therefore, cryo-polymerization is able to proceed with an acceptable rate even at relatively low temperatures. After thawing, the previous frozen medium melts, leaving pores among the framework of the polymeric pore-wall. Usually, cryo-polymerization is performed in water, and the temperature should be −15 °C or lower. The cryo-polymerization of hydrophobic monomers should be performed in organic solvents of benzene, 1,4-dioxane [34] and acetic acid [36]. The higher freezing point of the above solvents reduces the energy consumption for freezing the medium, assisting in the preparation of cheap oil sorbents.

Herein, hydrophobic cryogels of poly(lauryl acrylate-*co*-ethylene glycol dimethacrylate) [P(LA-*co*-EGDMA)] are prepared through cryo-polymerization with 1,4-dioxane as the medium and porogen. The influences of the cryo-polymerization temperature, total monomer concentration and cross-linker content on the oil absorption performance are thoroughly investigated. This fabrication takes advantage of the cheap starting materials, facile and reproducible techniques and versatile modulation of the sorbent structure, making it an attractive route to develop oil-sorbents with comprehensive performance.

## 2. Experimental Section

### 2.1. Materials

Lauryl acrylate (LA) as the hydrophobic monomer and ethylene glycol dimethacrylate (EGDMA) as the cross-linker were purchased from Aladdin (Shanghai, China) and passed through a basic Al_2_O_3_ (Sinopharm Chemical Regent, Shanghai, China) column to remove the inhibitors. Benzoyl peroxide (BPO, Aladdin) as the oxidative initiator was purified by re-crystallization with chloroform/methanol. *N*, *N*-Dimethylaniline (DMA, Aladdin) was used as the reductive initiator. 1, 4-Dioxane, toluene, benzene, petroleum ether (60–90 °C) and ethanol were obtained from Sinopharm Chemical Reagent. Gasoline oil (92^#^, SinoPEC, China) and lubricating oil (APISG, Zunke, China) were purchase from the local supermarket. The hydrophobic dye of Sudan III was also purchased from Aladdin.

### 2.2. Preparation of Hydrophobic Cryogels through Cryo-Polymerization in Dioxane

The hydrophobic cryogels of P(LA-*co*-EGDMA) were synthesized through free radical cryo-polymerization in a way similar to that in [36]. Information regarding different cryo-polymerizations is presented in Table 1, and a typical experiment is present below. After LA (1.58 mL, 1.40 g, 5.8 mmol) and EGDMA (0.033 mL, 35 mg, 0.2 mmol) were dissolved in 1,4-dioxane (6.38 mL), DMA in 1,4-dioxane (0.122 mol/L, 1.0 mL) was added. After being purged with nitrogen at room temperature for 5 min to remove possible oxygen, BPO in 1,4-dioxane (0.122 mol/L, 1.0 mL) was charged under magnetic stirring. Then, the final mixture was transferred into a plastic tube (diameter: 15 mm) and kept in a cryo-thermostat chamber (DR502, Shanghai Jianheng, China) at −18 °C. After the mixture was completely frozen, the chamber temperature was increased to 2 °C and stood for 20 h to complete the cryo-polymerization. The obtained cryogel of PLA-EGDMA with 3 mol% of EGDMA in all monomers was thawed at room temperature and washed with 1,4-dioxane by immersing the cryogel to remove the unreacted monomers. Afterwards, the cryogel was dried through freeze-drying and stored for further experiments. Based on the weight of dried cryogel, the monomer conversion was estimated.

A similar procedure was adopted for the preparation of other cryogels at the same initiator concentration ([BPO] = [DMA] = 12.2 mmol/L). The total monomer concentration ([M]_0_ = 0.4, 0.5 and 0.6 mol/L), molar percentage of EGDMA in all monomers (*f*_EGDMA_ = 3.0, 5.0 and 7.0 mol%) and cryo-polymerization temperature (*T*_cp_ = 4, 2 and 0 °C) were varied.

### 2.3. Determination of Oil Absorption Capacity

The swelling ratio (*SR*) was determined gravimetrically at room temperature according to the following Equation:(1)$$SR=\frac{W}{W_{0}}$$
where *W* and *W*_0_ are the weight of the swollen and dried cryogel, respectively.

For this determination, the dried cryogel (about 1.0 g) was soaked in the tested organic solvent or oil (50.0 mL) at room temperature. At different intervals, the swollen cryogel was taken out, wiped with toilet paper, and weighed. The procedure above was repeated until the weight of the swollen cryogel did not change. The swelling ratio at saturated absorption is used to obtain the absorption capacity as (absorption capacity = *SR* −1).

The ability for the obtained hydrophobic cryogels to separate an oil–water mixture was also studied. Thus, toluene containing a little amount of Sudan III was mixed with city water at a volume ratio of 1:3 (toluene/water), and a hydrophobic cryogel with enough mass was put inside. A digital camera was used to follow the oil–water separation.

### 2.4. Determination of Gel Fraction through Repeating Absorption

The swollen cryogel was recycled in two ways: (1) direct lyophilization, to remove volatile solvents such as benzene and gasoline oil; (2) replacing involatile solvent or oil with petroleum ether, and then lyophilization to remove the latter. The lyophilization of the cryogels, also known as freezing dryness, was performed using a freezing dryer (FDU-1110, Tokyo Rikakikai, Japan) by keeping the cryogels in the chamber at −20 °C under high vacuum (50 Pa) for 48 h to obtain the dried cryogels without any solvent. As for the cryogels which had absorbed involatile solvent or oil, they were immersed in petroleum ether (50 mL for 1.0 g of dried cryogel) for 24 h, during which petroleum ether was renewed every 2 h, until all of the involatile solvent or oil was completely replaced with petroleum ether. The recycled cryogel was then re-weighed to check the mass loss, and its absorption was repeated for the next cycles. In the case of petroleum ether absorption, the final weight (*W*_g_) of the dried cryogels at the fifth cycle was used to calculate the gel fraction (*GF* = *W*_g_/*W*_0_) of the cryo-polymerization product. Through this repeating absorption, the reusability of the cryogel to absorb the oil/organic solvent was assessed.

### 2.5. Instruments

Fourier transform infrared (FTIR) spectroscopy was performed on a Nicolet 6700 instrument (Thermo Fisher, USA) in KBr pellets with a wavenumber range of 4000–400 cm^−1^, scanning frequency of 4 cm^−1^ and 16 scanning times. Scanning electron microscopy (SEM, JSM 6700F, Jeol Japan) was used to observe cryogel morphology at an accelerating voltage of 10 kV after spluttering the samples with gold for 5 min under vacuum. A nitrogen adsorption–desorption experiment was also used to estimate the macro-porous interior of cryogels using a Belsorp-HP surface area analyzer (ASAP 2460 2.02 Bel, Japan) at 77 K. Prior to measurement, the sample was degassed at 120 °C under a high vacuum for 4 h. The static contact angle against water was measured on an optical contact angle meter (Solon Shanghai Tech Science) at ambient temperature. Five measurements at different locations of cryogels were averaged for the values of contact angles. Thermo-gravimetric analysis (TGA) was performed on a (TA Q5000IR, USA) instrument at a heating rate of 10 °C/min under nitrogen flow (100 mL/min). About 2 mg of dried samples was used, and the temperature range was set from 50 to 600 °C.

## 3. Results and Discussion

### 3.1. Synthesis of Macro-Porous Hydrophobic Cryogels

Through the conventional polymerization of hydrophobic monomers in solution, gels with micro-pores are usually obtained and have been widely used as oil-sorbents. To acquire other kinds of oil-sorbents with a different porosity morphology, macro-porous hydrophobic cryogels of P(LA-*co*-EGDMA) were prepared through the cryo-polymerization of lauryl acrylate with an EGDMA cross-linker, as shown in Scheme 1. Since low temperatures below the freezing point of the polymerization mixture are critical to cryo-polymerization, more energy should be consumed compared with the common polymerization at increasing temperatures. However, as for the cryo-polymerization of oil-soluble monomers, organic solvents such as 1,4-dioxane [34], acetic acid [36] and benzene instead of water should be used as the polymerization medium. The relative higher freezing point allows the cryo-polymerization of oil-soluble monomers to be achieved at a positive temperature. Compared with the freezing point of acetic acid (16.6 °C), 1,4-dioxane has a lower value (11.8 °C), but it is environmentally friendly. Therefore, the present cryo-polymerizations were carried out in a similar manner to the previous report [36], with 1,4-dioxane as the reaction medium instead of acetic acid. It is also found that the present cryo-polymerizations of LA and EGDMA in 1, 4-dioxane took 20 h to complete, but those of lauryl methacrylate and divinylbenzene in acetic acid were reported to require 48 h [36]. Thus, 1,4-dioxane was preferred over acetic acid considering the cryo-polymerization duration and green chemistry.

For the current cryo-polymerizations, all the starting mixtures were frozen to −18 °C to ensure their complete freezing. Then, the mixture temperature was raised to 0, 2 and 4 °C, respectively. At those temperatures, all the were kept bulkily frozen and cryo-polymerizations occurred, resulting in P(LA-*co*-EGDMA) cryogels. The effects of the cryo-polymerization temperature, total monomer concentration and molar percent of EGDMA in all monomers on the formation and oil absorption capacity of cryogels were investigated symmetrically.

Firstly, cryo-polymerizations were carried out at 2 °C with different [M]_0_ and *f*_EGDMA_. As for [M]_0_ = 0.4 mol/L, the product of the liquid mixture was obtained at three *f*_EGDMA_ values. After checking the left monomer in the obtained mixture by adding iodine solution, it was found that the purple–brown color of the iodine solution quickly disappeared, indicating that the monomer conversion was fairly low. As for [M]_0_ = 0.5 mol/L, elastic cryogels were obtained for three *f*_EGDMA_ cases, and there was little remaining monomer according to the above test. However, the gel fraction was not larger than 80%, even at the highest *f*_EGDMA_ value. It was found that [M]_0_ = 0.6 mol/L was the appropriate choice, since the monomer conversion reached 100% and the gel fraction exceeded 95%. Thus, the cryo-polymerizations of [M]_0_ = 0.6 mol/L were selected in the further investigation, and the related information is summarized in Table 1.

To confirm the occurrence of monomer polymerization, FTIR characterization was performed. Typical FTIR spectra are shown in Figure 1A. The absorbance signals at 1739 and 1261 cm^−1^ are attributed to the C=O stretching vibration of the ester carbonyl group [37] and the C–O stretching vibration of the ester group [36], respectively. The broad absorbance signal in 2600–3000 cm^−1^ is assigned to C–H stretching vibration [37]. The absorbance signal at 1468 cm^−1^ comes from the scissor vibration and asymmetric deformation vibration of the methylene group, while that at 1377 cm^−1^ originates from the symmetric deformation vibration of the methyl group. The absorbance signal at 721 cm^−1^ is characteristic of the in-plane rocking vibration of the multi-methylene group (CH_2_)*_n_*, definitely supporting the polymerization of LA with a long alkyl group.

Unfortunately, FTIR analysis could not show the composition variation of cryogels exactly since the initial molar percent of EGDMA was changed slightly. However, according to the saturated swelling degree and gel fraction, the crosslinking degree could be assessed and the incorporation amount of EGDMA into the cryogels could be estimated. The lower EGDMA mol% in the monomer mixture, the lower the gel fraction of obtained cryogels, as shown in Table 1. The result supported the common-sense idea that a higher cross-linker amount leads to a higher crosslinking degree.

It has been reported that the cryo-polymerization temperature has considerable impact on the polymerization rate and molecular weight of polymers [38]. In our case, the gel fraction and monomer conversion were the highest at 2 °C, suggesting that the cryo-polymerization rate at this temperature would be the highest. The interplay between the cryo-concentration effect leading to the rate increasing and the lessening temperature leading to the rate decrease is attributed to this consequence.

The thermal stability of cryogels was evaluated though the TGA technique, and thermograms of dried C-63p2, C-65p2 and C-67p2 cryogels made from different EGDMA mol% are shown in Figure 1B. The cryogels with different crosslinking degrees exhibit quite similar thermal degradation behavior except for the final weight loss. Up to 220 °C, only a little weight loss is observed. A sharp weight loss starts at about 280 °C and finishes at 450 °C, resulting from polymer thermal degradation. However, the starting temperature of the sharp weight loss for the C-67p2 cryogel is a little higher than those of other cryogels, indicating that C-67p2 cryogel is more thermally stable. TGA results suggest that the fabricated cryogels could stand temperatures as high as 200 °C. The performance of cryogels as oil absorbent at high temperatures has also been studied, as shown later.

### 3.2. Morphology and Porosity of Cryogels

During the cryo-polymerization, the medium of 1,4-dioxane was mostly frozen, and monomers were polymerized in the unfrozen liquid micro-phase (UFLMP). After the monomers were completely converted, crosslinked polymer network was formed in UFLMP to act as the framework of the cryogel; i.e., the pore-walls. The frozen medium of 1,4-dioxane, when thawed, left its occupied space as the inter-connected pore of the cryogel.

The inner morphology of the as-prepared cryogels was observed under SEM, as shown in Figure 2. Comparing SEM images of cryogels prepared at different temperatures, it was found that the pores of cryogels prepared at 2 °C were mostly open and large in size. On the contrary, the pore tunnels of cryogels prepared at 4 °C were somewhat blocked by the pore-walls, and their average sectional size was relatively smaller than that at 2 °C.

It is reasonably assumed that, upon raising the temperature of the polymerization mixture from −18 °C, some solvent crystal would melt into liquid 1,4-dioxane, and the volume fraction of the unfrozen liquid microphase (UFLMP) would increase. UFLMP, as the dispersed domain in the polymerization mixture, would get larger and become inter-connected at a certain temperature for cryo-polymerization. In the current case, when the temperature for the cryo-polymerization was pre-set to 2 °C, UFLMP existed mainly as dispersed domains with a smaller volume fraction. Therefore, after cryo-polymerization and thawing, the previous UFLMP domains transformed into the pore-walls, likely being more separated. The frozen medium was converted into the open pores to a greater extent. When the temperature for cryo-polymerization was pre-set to 4 °C, several UFLMP domains fused into a larger one, leading to the blockage of several pore tunnels of the obtained cryogels.

As shown in Figure 2, the pore sectional size of cryogels is in the range of 20~200 μm, and the pores are inter-connected to different extents, which is important for those cryogels to exhibit good oil absorption performance. Comparing the inner morphology of cryogels made from different *f*_EGDMA_ values, it is found that the pore size seems smaller and the pore looks more exposed with an increased *f*_EGDMA_. At higher *f*_EGDMA_ values, the apparent molecular weight and crosslinking degree of the produced polymer are higher, leading to a lower movement ability of polymer chains and chain segments. Thus, UFLMP domains would have greater difficulty growing inter-connected, resulting in an inner morphology with exposed macro-pores. Since the pores are irregular in shape and poly-dispersed in diameter, the quantitative identification of pore size is difficult based on SEM photos. Thus, a nitrogen absorption–desorption examination was further used to confirm the macro-porosity of different cryogels. The nitrogen adsorption–desorption technique is widely adopted to determine the porosity and total surface area of micro/meso-porous materials [39], but it is still questionable for its usage in macro-porous and soft materials [40]. As shown in Appendix A, the nitrogen adsorption–desorption isotherms of four cryogels are characteristic of a Type II curve with a very small hysteresis cycle, suggesting that all the cryogels are macro-porous on the whole [41]. The presence of hysteresis loops indicates the co-existence of micro-pores in the cryogels.

### 3.3. Hydrophobicity of PLA-EGDMA Cryogels

P(LA-*co*-EGDMA) cryogels are intrinsically hydrophobic due to the presence of long lauryl groups. The contact angle of as-prepared cryogels against water was determined, and typical camera photos are shown in Figure 3. Due to the irregular surface of cryogels, the measured contact angle might not reflect the real hydrophobicity of cryogel polymers themselves. However, the averaged contact angles of different cryogels against water are found to be in the range of 115~130°. For example, the values are 128°, 118° and 120° for cryogels of C-63p2, C-65p2 and C-67p2, respectively. Contact angle analysis confirms that the as-prepared cryogels are highly hydrophobic in nature.

### 3.4. Influences of Different Parameters on the Absorption Capacity of Cryogels

Since all of the three cryogels prepared at 2 °C had a reliable gel fraction (not less than 95%), they were chosen for a thorough investigation of the oil absorption performance of P(LA-*co*-EGDMA) cryogels with inter-connected macro-porosity. Different organic solvents (petroleum ether, benzene and toluene) and oils (gasoline and lubricating oil) were used as the absorbed medium. The results are illustrated in Figure 4.

Firstly, the influence of the crosslinking degree of cryogels on absorption capacity is considered. As demonstrated in Figure 4A, the absorption capacities of C-63p2, C-65p2 and C-67p2 cryogels with different *f*_EGDMA_ values decrease with the crosslinking degree for all three hydrophobic absorbates. Higher *f*_EGDMA_ values in the initial monomers result in a higher crosslinking degree in the obtained cryogel, as well as the decrease of the swelling degree according to the general knowledge of crosslinked polymers. As for macro-porous cryogels, the absorbed solvent molecules are not only distributed in the polymer network of the cryogel pore-wall but also held in the cryogel macro-pores. As indicated by SEM, C-67p2 with a higher crosslinking degree has a larger pore-size. However, its absorption capacity is a little lower compared with the other two cryogels. This is possibly caused by the higher crosslinking degree of the polymer restricting the expansion of the cryogel framework, limiting the pore expanding in volume during the absorption of the oil/solvent. Thus, it is assumed that the crosslinking degree, rather than pore size/volume, would be the dominant factor affecting the saturated swelling degree; i.e., the absorption capacity of macroporous cryogels.

The absorption capacity of the obtained cryogels also depends on the absorbed substance. As shown in Figure 4B, the C-63p2 cryogel has different absorption capacities which varied by solvent/oil type. Obviously, the absorption capacities for the polar solvent of methanol are significantly low compared with others, due to the large dissimilarity of its chemical structure with P(LA-*co*-EGDMA) polymer. The solubility parameter (*δ*) is commonly used to evaluate the swelling degree of cross-linked polymers in solvent. For a coupled non-polar polymer and non-polar solvent without volume change during swelling, the mixing enthalpy (∆*H*_mix_) is given as
(2)$$∆H_{mix}=V_{M}{\left(\delta_{p}-\delta_{s}\right)}^{2}\varphi_{p}\varphi_{s}$$
where *δ*_p_ and *δ*_s_ are the solubility parameters of the polymer and solvent, respectively. *φ*_p_ and *φ*_s_ are the volume fractions of the polymer and solvent, respectively. *V*_M_ is the total volume.

The solubility parameter of pure poly(lauryl acrylate) is 16.7 J^1/2^/cm^3/2^, approximately considered as that of P(LA-*co*-EGDMA) in the current case. The solubility parameters of toluene and benzene are 18.2 and 18.7 J^1/2^/cm^3/2^, respectively. Petroleum ether, gasoline oil and lubricating oil are the mixtures of different aliphatic hydrocarbons. The solubility parameter of alkanes increases with carbon number and is lower than that of PLA-EGDMA, such as *n*-butane (13.5 J^1/2^/cm^3/2^), *n*-pentane (14.4 J^1/2^/cm^3/2^), *n*-hexane (14.9 J^1/2^/cm^3/2^) and *n*-octane (15.4 J^1/2^/cm^3/2^). Thus, it reasonable to deduce that the solubility for petroleum ether, gasoline oil and lubricating oil is close to that of PLA-EGDMA, causing the lowest value of the absorption capacity of petroleum ether.

Besides this, the density of the solvent/oil is important regarding the effect of the absorption capacity because the volume is the primary aspect which influences how the porous absorbent holds the absorbate. Since toluene (0.866 g/mL) and benzene (0.876 g/mL) have a similar density as well as a similar chemical structure, their absorption capacities are very close. The density of petroleum ether is about 0.65 g/mL, while those of gasoline oil and lubricating oil are 0.70~0.78 and 0.75~0.95 g/mL, respectively. Taking the density into consideration, the absorption capacity for petroleum ether is still the lowest.

Furthermore, the absorption rate for a cryogel to capture the solvent/oil was also verified. The dependence of the swelling ratio on time for the C-63p2 cryogel to absorb solvent/oil is shown in Figure 4C. Except for lubricating oil, the swelling ratio for the others reaches a saturated value within 3 min, including the hydrophilic solvent of methanol. The features of macro-porosity and the inter-connected pathway of cryogels should contribute to the rapid absorption. As for lubricating oil, it takes about 90~120 min to reach a saturated swelling state. This observed difference in absorption kinetics could be caused by the viscosity variation of solvent/oil. The viscosity of lubricating oil is usually much higher than organic solvents and gasoline oil, whose viscosities are less than 1 mPa·s. A high viscosity absolutely limits the diffusion of lubricating oil into the macro-porous cryogel; therefore, it takes a longer time for the cryogels to trap lubricating oil in an equilibrium state.

P(LG-*co*-EGDMA) cryogels prepared at 4 °C and [M]_0_ = 0.6 mol/L were also tested to disclose the effect of cryo-polymerization temperature on absorption performance, compared in Figure 4D and Figure 4B. Considering the same *f*_EGDMA_ and solvent/oil type, C-6*p4 cryogels should have quite similar absorption properties in comparison with C-6*p2 cryogels; however, the absolute absorption capacities of C-6*p4 cryogels are higher than those of their counterparts prepared at 2 °C. Herein, C-63p4 cryogel is exemplified as indicated in Figure 4D. The absorption capacities of the C-63p4 cryogel for different solvent/oil combinations are about 30% higher. As suggested by the morphology observation and pore analysis, C-6*p4 cryogels prepared at 4 °C have a smaller pore size because of the larger UFLMP volume. Keeping in mind that UFLMP is the locus of cryo-polymerization and converts into the pore-wall of cryogels, the packing and entanglement of the polymer chains is looser in the C-63p4 cryogel, making the pore wall expand with less trouble and allowing the pore to hold a greater solvent/oil volume.

To check the removal of floating oil with P(LA-*co*-EGDMA) cryogels, toluene (3 mL) colored with Sudan III was dropped onto water (10 mL) in a glass beaker. As shown in Figure 5A, the oil–water interface is clearly distinguished at a scale line of 10. A piece of cylindrical C-63p2 cryogel was put into the beaker, and oil absorption occurred. After 3 min, the cylindrical cryogel was taken out. As shown in Figure 5B, no trace of red oil is observed in the beaker, while the cylindrical cryogel becomes red and larger in size. The top-surface of the water remains at the same height, indicating that water has hardly been absorbed. As for benzene, petroleum ether and gasoline oil, the floating solvent/oil on the water surface can be completely removed within 3 min (see the video in the Appendix A). However, the complete removal of the floating lubricating oil takes a long time. These results demonstrate that the macro-porous cryogels are very useful to remove floating oil with a low viscosity.

### 3.5. Repeating Absorption and High-Temperature Tolerance of Hydrophobic Cryogels

The reusability of oil absorbents is very important for practical applications. Thus, after the careful removal of an oil or solvent absorbed through direct or indirect lyophilization, the recycled cryogels undergo the next absorption cycle. The lyophilization was performed under a very low temperature, ensuring the maintenance of the cryogel porosity morphology, as the cryogel became rigid to avoid the variation in its shape and volume. Replacing involatile solvent or oil with petroleum ether prompted the lyophilization of related cryogels. Although those recycling routes are different from the common squeezing method, they can completely remove the absorbed oil or solvent to obtain the exact mass of recycled oil-sorbent cryogels. In addition, the possible mass loss of cryogels could be detected. Figure 6A exhibits the variation of the absorption capacities of the C-63p2 cryogel for benzene and gasoline oil with the absorption cycle. After each absorption cycle, the cryogels were attentively recycled in complete dryness with the necessary replacement of high-boiling-point oil with a volatile solvent. Then, the recycled cryogels were used for further study. Within 10 absorption cycles, C-63p2 cryogel was observed to maintain a similar absorption capacity to benzene (about 14.0 g/g) and gasoline oil (about 11.5 g/g). The inner morphology of C-63p2 cryogel at the tenth cycle was also re-examined. As shown in Figure 2G, no obvious change of the inner morphology can be distinguished. In addition, the cryogels retain their apparent shape, and no breakage is observed within 10 absorption cycles. Based on the results, the P(LA-*co*-EGDMA) cryogels have satisfactory reusability for practical application in floating oil removal.

Usually, floating oil removal and oil absorption encounters high-temperature conditions. The TGA results shown in Figure 1B indicate no obvious weight loss up to 220 °C. To evaluate the high-temperature tolerance of P(LA-*co*-EGDMA) cryogels, they were kept at different temperatures (25, 50, 100, 150 and 200 °C) under vacuum for 3 h. After this treatment, the inner morphology and absorption capacity were re-tested. Figure 6B shows the dependence of the absorption capacity of the coupled C-63p4/benzene on treatment temperature, indicating a small decrease in absorption capacity at lower temperatures. As suggested in Figure 2H and Appendix A (50, 100 and 150 °C), the inner morphology of the treated cryogels hardly changed. However, when the treatment temperature was set to 200 °C, the absorption capacity obviously declined. Shown in Appendix A (200 °C) is one SEM photo of the C-63p4 cryogel after thermal treatment at 200 °C for 3 h. Less-open pores are observed. This phenomenon might be caused by the chain adjustment and polymer fusion during heating. Furthermore, the storage stability of the as-prepared cryogels as oil-sorbents was also checked. After being stored in a desiccator for six months, the cryogels exhibited an unchanged absorption capacity. For example, the C-63p4 cryogel had an absorption capacity of about 20 g/g just after its preparation, and this absorption capacity value was retained after the six-month storage of the C-63p4 cryogel. These results confirm the high storage stability of P(LA-*co*-EGDMA) cryogels, making them valuable candidates for oil-absorbents and as floating oil removal materials.

In the present research, the overall oil absorption performance of P(LA-*co*-EGDMA) cryogels is ascertained to be quite good and adjustable by several preparation parameters. The absorption capacities for organic solvents and oils are commonly larger than 9 g/g, even reaching 20 g/g. Table 2 summarizes the absorption capacity of other oil absorbents reported in the literature, taking the materials and fabrication methods into account.

The presented P(LA-*co*-EGDMA) cryogels have a somewhat higher absorption capacity than most of the reported absorbents. Poly(butyl methacrylate)/organo-attapulgite (PBMA/OA) nano-composites [42] and chitin sponge [10] exhibit a higher oil absorption capacity than P(LA-*co*-EGDMA) cryogels, but the chemical vapor deposition of methyltrichlorosilane should be performed to increase their oleophilicity. Reaching saturated swelling should take 80, 60 and 40 min for benzene, gasoline oil and chloroform, respectively. In addition, their absorption capacity obviously decreases at the second absorption cycle, then remains unchanged. As reported in [12], Jiang et al. prepared magnetic TiO_2_ foams with a commercial polyethylene shock absorption foam as the support. Those foams possess an extremely high absorption capacity (up to 60 g/g) and excellent reusability (up to 80 cycles), which is claimed to be caused by the presence of TiO_2_ nanoparticles. The whole preparation route includes the uploading of oleic acid-coated Fe_3_O_4_ nanoparticles, the uptake of TiO_2_ nanoparticles and treatment with methyltrichlorosilane in ethanol, which is quite complicated.

In the present work, the preparation of inter-connected macroporous P(LA-*co*-EGDMA) cryogels is facile, low-cost and reproducible, while their oil absorption performance is comprehensively better than other methods. Therefore, cryogels from cryo-polymerization open an alternative route for the development of oil-sorbents and floating oil removal materials.

## 4. Conclusions

Hydrophobic P(LA-*co*-EGDMA) cryogels with an inter-connected macro-porosity have been straightforwardly prepared through the cryo-polymerization of lauryl acrylate and ethylene glycol methacrylate in 1,4-dioxane at different low temperatures. The influences of preparation parameters such as the total monomer concentration, cross-linker molar percent, and cryo-polymerization temperature on the inner morphology as well as the oil-absorption performance have been investigated symmetrically. A high monomer conversion and gel fraction near to 100% are achieved at 2 °C and [M]_0_ = 0.6 M, with an FTIR confirmation of the successful cryo-polymerization. The macro-porous structure of the as-prepared cryogels was affected by the cryo-polymerization temperature along with the crosslinking degree and monomer concentration. A higher temperature, lower cross-linker amount and lower monomer concentration of cryo-polymerization yielded cryogels with larger and more open-cell macro-pores, but the nature of the pore wall might have an important effect on absorption. These results absolutely demonstrate that the cryogel structures, including the chemical composition, crosslinking degree, and pore morphology, could be modulated through various fabrication parameters. With characteristic features such as hydrophobicity and open-cell macro-pores, the cryogels possessed an absorption capacity as high as 20–21 g/g for different organic solvents and oils. Additionally, the cryogels exhibited a rapid upload of organic solvents and oils to reach the maximum absorption within 3 min except for highly-viscous lubricating oil. Furthermore, the cryogels are easily recycled without cryogel mass loss and can be re-used without lessening absorption capacity for at least 10 cycles. Their high-temperature tolerance is fairly high, without any decrease of absorption capacity after heating treatment up to 150 °C for 3 h. The convenient and low-cost fabrication, facile modulation of the structure and satisfactory performance of P(LA-*co*-EGDMA) cryogels open a hopeful route for the development of novel oil-absorbents.

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
