# Peer review of "Macroporous Oil-Sorbents with a High Absorption Capacity and High-Temperature Tolerance Prepared Through Cryo-Polymerization"

_polymers, 2019, doi:10.3390/polym11101620_

Round 1
Reviewer 1 Report
The manuscript entitled “Macroporous Oil-sorbents with High Absorption Capacity and High-temperature Tolerance Prepared through Cryo-polymerization“ presents the way of synthesis and absorption properties of macroporous polymeric sorbents. Very extensive laboratory work was done to prepare the described materials and test their properties. However, Results and Discussion part need thorough revision and correction.
Lines 140-1: “The swollen cryogel was recycled by direct lyophilization or indirect lyophilization after replacing the high boiling point medium with petroleum ether.”
Comment: the fragment is unclear and need additional explanation, how the lyophilization was performed, what was temperature and pressure?
Lines 162-3: “Through conventional polymerization of hydrophobic monomers in solution, gels with micro-162 porous are usually obtained, leading to low absorption capacity of the resultant materials”
Comment: microporous materials are used as adsorbents and in order to be effective in this process they must have good (ad)sorption properties.
Line 203: it should be FTIR instead of FTIT (dotted lines in the plot of figure 1 are hardly legible)
Lines 215-6: The result defended the common sense that higher cross-linker amount leads to higher crosslinking degree.
Remark: this statement is true for vinyl cross-linkers like e.g. divinylbenzene. If the Authors obtained different results they should try to explain this phenomenon.
Lines 224-9: Figure B1 shows only one TG result.
A reader cannot compare it with those for the other materials. General rule is that the higher crosslinking, the better thermal resistance. Very useful can be following publications:
Levchik, G.F.; Si, K.; Levchik, S. V.; Camino, G.; Wilkie, C. a. The correlation between cross-linking and thermal stability: Cross-linked polystyrenes and polymethacrylates. Polym. Degrad. Stab. 1999, 65, 395–403, doi:10.1016/S0141-3910(99)00028-2. Wilkie, C. a. TGA/FTIR: an extremely useful technique for studying polymer degradation. Polym. Degrad. Stab. 1999, 66, 301–306, doi:10.1016/S0141-3910(99)00054-3.It’s a pity that only one result was presented. Additionally, TG allows to evaluate degree of crosslinking, and DSC can reveal/confirm the presence of unreacted vinyl bonds.
Line 267 and 384 : Figures S1 and S2 were not submitted.
Line 269: occurrence of hysteresis loop is characteristic for mesoporous materials. Please see: Pure & Appl. Chem., Vol. 57, No. 4, pp. 603—619, 1985.
Lines 293-4:” As indicated by SEM, C-293 67p2 with higher crosslinking degree has larger pore-size/volume.”
Remark: pore volume cannot be evaluated on the base of SEM images. SEM allow only to “see” the size of pore entrances. Pore volume can be obtained by e.g. porosimetry or SAXS.
Legend of Figure 4 B: there is methano instead of methanol
Line 365: how the removal was carried out?
Lines 389-391: “…C-63p4 cryogel kept the absorption capacity of about 21 g/g just after its preparation and after six-month storage. Those results confirm the high storage stability of P(LA-co-EGDMA) cryogels, making them the valuable candidates for oil-absorbents and floating-oil removal materials. “
Remark: this fragment is unclear. What were conditions of storage of the material? Was it dry or with an absorbate?
Page 12: Table should have no.2
Author Response
Reviewer 1
The manuscript entitled “Macroporous Oil-sorbents with High Absorption Capacity and High-temperature Tolerance Prepared through Cryo-polymerization presents the way of synthesis and absorption properties of macroporous polymeric sorbents. Very extensive laboratory work was done to prepare the described materials and test their properties. However, Results and Discussion part need thorough revision and correction.
Response: Thanks for this suggestion. We have checked “Results and Discussion” section and made necessary revision and correction, based both on the comments from all the reviewers and our checking. The changes are highlighted in blue color.
Lines 140-1: “The swollen cryogel was recycled by direct lyophilization or indirect lyophilization after replacing the high boiling point medium with petroleum ether.”
Comment: the fragment is unclear and need additional explanation, how the lyophilization was performed, what was temperature and pressure?
Response: Thanks for this suggestion. Indeed, the previous description was not clear. Thus, it has been changed into “The swollen cryogel was recycled through two routes: 1) direct lyophilization to remove volatile solvents such as benzene and gasoline oil; 2) replacing non-volatile solvent or oil with petroleum ether and then lyophilization to remove the latter.” Additional sentences have been added at the beginning of Subsection 3.5 of the revised manuscript to explain why these recycle routes are used.
Lines 162-3: “Through conventional polymerization of hydrophobic monomers in solution, gels with microporous are usually obtained, leading to low absorption capacity of the resultant materials”.
Comment: microporous materials are used as adsorbents and in order to be effective in this process they must have good (ad)sorption properties.
Response: It is interesting comment. Indeed, microporous materials are used as adsorbents with good (ad)sorption properties. However, these properties are dependent on various structure parameters, including chemical structure and porous morphology. In this manuscript, conventional gels did exhibit lower absorption capacity. As well, the conventional gel took a long time to reach to their swelling equilibrium while cryogels only took 1-3 minutes. Thus, a video is presented in Supporting Information.
Line 203: it should be FTIR instead of FTIT (dotted lines in the plot of figure 1 are hardly legible)
Response: Great thanks for those comments. The type-error has been rectified. The dotted lines in Figure 1A have been changed into solid lines with different colors.
Lines 215-6: The result defended the common sense that higher cross-linker amount leads to higher crosslinking degree.
Remark: this statement is true for vinyl cross-linkers like e.g. divinylbenzene. If the Authors obtained different results they should try to explain this phenomenon.
Response: Thanks for this comment. In our present work, we did not obtain different results from the common sense that higher cross-linker amount leads to higher crosslinking degree. We also further investigated the effect of cross-linking degree on absorption capacity and micro-structure. With the enhancement of cross-linker concentration, small pores size and low absorption capacity observed.
Lines 224-9: Figure B1 shows only one TG result.
A reader cannot compare it with those for the other materials. General rule is that the higher crosslinking, the better thermal resistance. Very useful can be following publications: Levchik, G.F.; Si, K.; Levchik, S. V.; Camino, G.; Wilkie, C. a. The correlation between cross-linking and thermal stability: Cross-linked polystyrenes and polymethacrylates. Polym. Degrad. Stab. 1999, 65, 395–403, doi:10.1016/S0141-3910(99)00028-2. Wilkie, C. a. TGA/FTIR: an extremely useful technique for studying polymer degradation. Polym. Degrad. Stab. 1999, 66, 301–306, doi:10.1016/S0141-3910(99)00054-3.
It’s a pity that only one result was presented. Additionally, TG allows to evaluate degree of crosslinking, and DSC can reveal/confirm the presence of unreacted vinyl bonds.
Response: Great thanks for this comments. Actually, we did measured TGA thermograms of cryogels with different crosslinking degrees. However, in the range of current crosslinker monomer amount, the thermal degradation behaviors are quite similar except the starting temperature of sharp weight loss is a little different. In the revised manuscript, TGA thermograms of three cryogels have been offered and the result is briefly discussed.
Line 267 and 384: Figures S1 and S2 were not submitted.
Response: Thanks for the reviewer. Supporting Information with Figure S1 and S2 have been offered and uploaded at the submission of revised manuscript (Find in word file).
Line 269: occurrence of hysteresis loop is characteristic for mesoporous materials. Please see: Pure & Appl. Chem., Vol. 57, No. 4, pp. 603—619, 1985.
Response: Thanks for this comment and suggestion. We have cited this article in the revised manuscript and also referred to the other publications. In most of our cases, the hysteresis loops are very small, suggesting the vague meso-porous feature. As well, for cryogels and gels, there are the possible existence of hierarchical porous structure.
Lines 293-4:” As indicated by SEM, C-67p2 with higher crosslinking degree has larger pore-size/volume.”
Remark: pore volume cannot be evaluated on the base of SEM images. SEM allow only to “see” the size of pore entrances. Pore volume can be obtained by e.g. porosimetry or SAXS
Response: Great thanks and we agree to this comment. Thus, the word of “volume” is removed in the revised manuscript.
Legend of Figure 4 B: there is methano instead of methanol
Response: Thanks for this pointing-out. We have rectified this type-error of Figure 4B.
Line 365: how the removal was carried out?
Response: This issue is almost the same as the first one. In this revised manuscript, the detail description and related discussion have been offered, just below the title of Subsection 3.5.
Lines 389-391: “…C-63p4 cryogel kept the absorption capacity of about 21 g/g just after its preparation and after six-month storage. Those results confirm the high storage stability of P(LA-co-EGDMA) cryogels, making them the valuable candidates for oil-absorbents and floating-oil removal materials.
Remark: this fragment is unclear. What were conditions of storage of the material? Was it dry or with an absorbate?
Response: We agree to and are thankful for this comment. In the revised manuscript, the unclear sentence has been re-written.
Page 12: Table should have no.2
Response: Thanks for this comment. We have corrected this error.

Reviewer 2 Report
The current manuscript (ref. no.595513) deals with macroporous oil-sorbents prepared via cryo-polymerization technique. The sorbents were characterized by physical-chemical methods and showed good performance for oil spill cleanup. The content is appropriate in scope and level for the Journal. The manuscript is well written and structured. However, there are some aspects that need to be clarified.
1) In section 2.3, the swelling ratio (SR) is given by Eq.1 as [SR=W/W0]. But, on lines 133-134 the authors wrote that SR is considered (at saturation) as absorption capacity. Note that, SR is not identical with absorption capacity, because the formula for Absorption Capacity (S) is different, that is: [S = (W-W0) /W0] (this formula should be mentioned in the manuscript, and correct data in the manuscript considering this formula for the absorption capacity).
2) From Fig.2 (SEM images /cross-section) authors concluded that pores size range is of 20-200 microns. In our opinion, this is not a sufficient analysis. Histograms of pore size distribution should be developed by authors for each type of the produced material. To read pore size from SEM-image we recommend the open-source software ImageJ (https://imagej.net/ImageJ).
3) The restoring and recycling of sorbents by direct/indirect lyophilization method is not a common technique. It would be interesting if authors can report the recovery efficiency of spent sorbents by using simple-squeezing technique or centrifugation.
4) As regards Table 1 (comparison of sorbent materials), several important types of sorbents reported in the literature (i.e. clay-aerogel, polymeric nonwovens, electrospun fibers) were omitted; therefore these type of sorbents can be included in Table 1 for comparison, see for example references:
https://doi.org/10.1016/j.seppur.2014.06.059
https://doi.org/10.1016/j.polymertesting.2017.02.024
https://doi.org/10.1016/j.jtice.2016.11.005
Author Response
Reviewer 2
The current manuscript (ref. no.595513) deals with macroporous oil-sorbents prepared via cryo-polymerization technique. The sorbents were characterized by physical-chemical methods and showed good performance for oil spill cleanup. The content is appropriate in scope and level for the Journal. The manuscript is well written and structured. However, there are some aspects that need to be clarified.
In section 2.3, the swelling ratio (SR) is given by Eq.1 as [SR=W/W0]. But, on lines 133-134 the authors wrote that SR is considered (at saturation) as absorption capacity. Note that, SR is not identical with absorption capacity, because the formula for Absorption Capacity (S) is different, that is: [S = (W-W0) /W0] (this formula should be mentioned in the manuscript, and correct data in the manuscript considering this formula for the absorption capacity).
Response: Thanks for this comment. Thus, absorption capacity has been re-defined as suggested. The related data have been changed and the related figures have been re-drawn accordingly.
From Fig.2 (SEM images /cross-section) authors concluded that pores size range is of 20-200 microns. In our opinion, this is not a sufficient analysis. Histograms of pore size distribution should be developed by authors for each type of the produced material. To read pore size from SEM-image we recommend the open-source software ImageJ (https://imagej.net/ImageJ).
Response: Great thanks for this useful suggestion. We have obtained this software ImageJ and tried to apply it in the determination of average pore size from SEM images. However, we not confidently sure the reading results since we have not fully known this software.
The restoring and recycling of sorbents by direct/indirect lyophilization method is not a common technique. It would be interesting if authors can report the recovery efficiency of spent sorbents by using simple-squeezing technique or centrifugation.
Respond: Great thanks for this suggestion. In practice, oil sorbents are recycled by simple-squeezing technique or centrifugation. However, during our initial research work, we found that cryogel mass loss occurred for those with low crosslinking degrees upon swelling and repeating swellings. Low crosslinking degree leads to low gel fraction. Therefore, we prefer to report the results using direct/indirect lyophilization method to recover the cryogel oil-sorbents. In the revised manuscript, the related description has been added to explain it.
As regards Table 1 (comparison of sorbent materials), several important types of sorbents reported in the literature (i.e. clay-aerogel, polymeric nonwovens, electrospun fibers) were omitted; therefore, these type of sorbents can be included in Table 1 for comparison, see for example references:
https://doi.org/10.1016/j.seppur.2014.06.059; https://doi.org/10.1016/j.polymertesting.2017.02.024;
https://doi.org/10.1016/j.jtice.2016.11.005
Response: Thanks for this suggestion. New contents have been added into Table 2 of the revised manuscript with the update references.
Reviewer 3 Report
In my opinion, in the present form the manuscript (polymers-595513) entitled ‘Macroporous Oil-sorbents with High Absorption Capacity and High-temperature Tolerance Prepared through Cryo-polymerization’ described by Abdul Haleem, Jia-Yun Wang , Hui-Juan Li , Chuan-Shan Hu , Xi-Chuan Li , Wei-Dong He can be recommended for publication in the Polymers after major revision.
My recommendations to the Polymers manuscript are as follows:
The text is comprehensible.
Authors should add:
What is the repeatability / reproducibility of the method?
After reading the publication, I have doubts about the scientific novelty of the scientific research of the reviewed manuscript. Therefore, the Authors should clearly indicate what is the scientific novelty of their research.
Conclusions are too laconic. Please indicate clearly what is new with your manuscript (Conclusions) for the Polymers, especially in comparison to earlier of publication(s), e.g.,
Synthesis of penetrable poly(methacrylic acid-co-ethylene glycol dimethacrylate) microsphere and its HPLC application in protein separation
Xiao Sun, Jing Li, Li Xu
Talanta Volume 185, 1 August 2018, Pages 182-190
Abstract
In the present study, the narrow-dispersed penetrable poly(methacrylic acid-co-ethylene glycol dimethacrylate) (poly(MAA-co-EDMA)) microspheres were successfully synthesized based on the sacrificial support method. The poly(MAA-co-EDMA) microspheres mirrored the porous structure of the sacrificial support, i.e. penetrable silica, characteristic of copious mesopores and throughpores. In addition, they possessed large surface area, adjustable hydrophobicity and the cation-exchange ability. Owing to their multi functionalities, they were applied as chromatographic stationary phase to separate proteins in different separation modes, including reversed phase, hydrophobic interaction and weak cation exchange. Moreover, thanks to their throughpores, fast separation at low column backpressure could be achieved in these three modes. Both protein recovery and column stability were satisfactory. The penetrable poly(MAA-co-EDMA) microspheres were potential stationary phase matrix for fast protein separation.
Some references are omitted, e.g.,
Poly[hydroxyethyl acrylate-co-poly(ethylene glycol) diacrylate] Monolithic Column for Efficient Hydrophobic Interaction Chromatography of Proteins, Yuanyuan Li, H. Dennis Tolley, Milton L. Lee, Analytical Chemistry 81(22):9416-24 · October 2009, DOI: 10.1021/ac9020038
Abstract
Rigid poly[hydroxyethyl acrylate-co-poly(ethylene glycol) diacrylate] monoliths were synthesized inside 75 mum i.d. capillaries by one-step UV-initiated copolymerization using methanol and ethyl ether as porogens. The optimized monolithic column was evaluated for hydrophobic interaction chromatography (HIC) of standard proteins. Six proteins were separated within 20 min with high resolution using a 20 min elution gradient, resulting in a peak capacity of 54. The effect of gradient rate and initial salt concentration on the retention of proteins were investigated. Mass recovery was found to be greater than 96%, indicating the biocompatibility of this monolith. The monolith was mechanically stable and showed nearly no swelling or shrinking in different polarity solvents. The preparation of this in situ polymerized acrylate monolithic column was highly reproducible. The run-to-run and column-to-column reproducibilities were less than 2.0% relative standard deviation (RSD) on the basis of the retention times of protein standards. The performance of this monolithic column for HIC was comparable or superior to the performance of columns packed with small particles.
Page 9 of 15, Figure 4B:
Should be: methanol
Author Response
Reviewer 3
In my opinion, in the present form the manuscript (polymers-595513) entitled ‘Macroporous Oil-sorbents with High Absorption Capacity and High-temperature Tolerance Prepared through Cryo-polymerization’ described by Abdul Haleem, Jia-Yun Wang , Hui-Juan Li , Chuan-Shan Hu , Xi-Chuan Li , Wei-Dong He can be recommended for publication in the Polymers after major revision.
My recommendations to the Polymers manuscript are as follows:
The text is comprehensible.
Authors should add:
What is the repeatability / reproducibility of the method?
Response: Thanks for this suggestion. Indeed, the different cryogels have been prepared at the different time and the properties/performance has been achieved at almost the same level. As well, different characterizations and tests have been repeated. Thus, the related description has been added at the end of manuscript main body.
After reading the publication, I have doubts about the scientific novelty of the scientific research of the reviewed manuscript. Therefore, the Authors should clearly indicate what is the scientific novelty of their research.
Response: Great thanks for this suggestion. Although we have discussed the current research and new development in oil-sorbents, the manuscript had not clearly demonstrated the scientific novelty. Thus, at the end of Introduction in the revised manuscript, we emphasize this with more description.
Conclusions are too laconic. Please indicate clearly what is new with your manuscript (Conclusions) for the Polymers, especially in comparison to earlier of publication(s), e.g., Synthesis of penetrable poly(methacrylic acid-co-ethylene glycol dimethacrylate) microsphere and its HPLC application in protein separation; Xiao Sun, Jing Li, Li Xu Talanta Volume 185, 1 August 2018, Pages 182-190; Poly[hydroxyethyl acrylate-co-poly(ethylene glycol) diacrylate] Monolithic Column for Efficient Hydrophobic Interaction Chromatography of Proteins, Yuanyuan Li, H. Dennis Tolley, Milton L. Lee, Analytical Chemistry 81(22):9416-24 · October 2009, DOI: 10.1021/ac9020038
Response: Great thanks for this suggestion. Conclusion has been re-written with more emphasis on the significance of manuscript research.
Page 9 of 15, Figure 4B: Should be: methanol
Response: Thanks for this comment as the same as the above. The type-error has been rectified in Figure 4B of the revised manuscript.
Round 2
Reviewer 1 Report
Thank for all incorporated changes, but there are still fragments that demand clarification:
Lines 143-5 and 372-4: : despite the improvements, still nothing is known how the lyophilization was performed, what was temperature and pressure?
To readers who are not familiar with lyophilization method it is still unclear how the process of removal and recycling is made.
"Thus, after the careful removal (what does is mean? sublimation? what condition was applied for careful removal and what would be not careful, does they cause some kind of damages to the gel? ) of oil or solvent absorbed through direct or indirect lyophilization, the recycled cryogels underwent the next absorption cycle.
How the solvents were replaces? by dipping and rinsing or squeezing and re-absorption (like in a sponge?)?
Lines 166-7: no change was made in the text.
Paragraphs 3.2 and 3.4 information regarding morphology of the presented materials: no porosity evaluation cannot be made based on SEM images. SEM allows only to “see” the size of pore entrances. At given magnification only the entrances of macropores and the biggest mezopores are visible. Porous structure parameters such as pore volume, pore size distribution or histogram can be obtained on the basis of N2 sorption experiments. From the nitrogen sotption isotherms in supplement one can see that also small amount of micropores is present in the gel structure. The shape or size of histeresis loop have not direct impact on amount of mezopores.
Author Response
REVIEWER 1
Thank for all incorporated changes, but there are still fragments that demand clarification:
1) Lines 143-5 and 372-4: despite the improvements, still nothing is known how the lyophilization was performed, what was temperature and pressure?
To readers who are not familiar with lyophilization method it is still unclear how the process of removal and recycling is made.
"Thus, after the careful removal (what does it mean? sublimation? what condition was applied for careful removal and what would be not careful, does they cause some kind of damages to the gel?) of oil or solvent absorbed through direct or indirect lyophilization, the recycled cryogels underwent the next absorption cycle.
How the solvents were replaces? By dipping and rinsing or squeezing and re-absorption (like in a sponge?)?
Response:
We did not really follow what this issue of comment in the first run of reviewing. We just thought that “lyophilization”, i.e. freezing dryness, has been frequently reported in the literature and thought the running temperature and pressure would not be needed. At the present, we get to know that we ignored this important information for common readers to follow. Thus, we revised the concerning description, including the instrument, temperature and vacuum degree, in Subsection 2.4. The advantage of lyophilization such as maintaining the porosity morphology has been offered in Subsection 3.5 of the new manuscript version.
The “careful removal” means that: 1) the lyophilization was used to obtain the dry cryogels without any solvent, which ensured the maintenance of cryogel porosity morphology; 2) involatile solvent or oil absorbed by the cryogels was replaced with petroleum ether before the lyophilization, which ensure the complete removal during the lyophilization and the exact mass of re-cycled cryogels for the next absorption.
The information about the replacement of involatile solvent or oil has been offered in Subsection 2.4 of the new manuscript version.
All the new revision has been marked in brown red.
2) Lines 166-7: no change was made in the text.
Paragraphs 3.2 and 3.4 information regarding morphology of the presented materials: no porosity evaluation cannot be made based on SEM images. SEM allows only to “see” the size of pore entrances. At given magnification only the entrances of macropores and the biggest mesopores are visible. Porous structure parameters such as pore volume, pore size distribution or histogram can be obtained on the basis of N2 sorption experiments. From the nitrogen sorption isotherms in supplement one can see that also small amount of micropores is present in the gel structure. The shape or size of hysteresis loop have not direct impact on amount of mesopores.
Response:
We misunderstand this comment previously. In the new version of revised manuscript, we made the change for the mentioned sentence.
We are a little confused about “no porosity evaluation cannot be made based on SEM images”. By careful reading the whole content as well as that in the previous comments, we get to notice that our cryogels have hierarchical porosity morphology with macro-, meso- and micro-pores, which will help our future research. Great thanks for the reviewers. Thus, we have made the changes at the end of Subsection 3.2 of the new version of revised manuscript.
Reviewer 3 Report
In my opinion, in the present form the manuscript (polymers-595513) entitled ‘Macroporous Oil-sorbents with High Absorption Capacity and High-temperature Tolerance Prepared through Cryo-polymerization’ described by Abdul Haleem, Jia-Yun Wang , Hui-Juan Li , Chuan-Shan Hu , Xi-Chuan Li , Wei-Dong He can be recommended for publication in the Polymers.
Author Response
Reviewer 3
In my opinion, in the present form the manuscript (polymers-595513) entitled ‘Macroporous Oil-sorbents with High Absorption Capacity and High-temperature Tolerance Prepared through Cryo-polymerization’ described by Abdul Haleem, Jia-Yun Yang, Hui-Juan Li, Chuan-Shan Hu, Xi-Chuan Li, Wei-Dong He can be recommended for publication in the Polymers.
Response: There is not suggestion or advice in this comment. We are thankful for the reviewer approval and help.
Round 3
Reviewer 1 Report
Thank you for the effort put in to improving the article.
Now, it is obvious how the freeze-drying process was carried out.
The only remark concerns the last sentence in paragraph 3.2
"The presence of hysteresis loops indicates the co-existence of micro-pores in the cryogels."
it should be meso-pores instead of micro-pores.